# Granzyme B PET Imaging Enables Detection of CAR T-Cell Therapy Response in a Human Melanoma Mouse Model

**DOI:** 10.3390/diagnostics15233058

**Published:** 2025-11-30

**Authors:** Priska Summer, Niklas Bulmer, Suma Prabhu, Naomi Gallon, Rebecca C. Larson, Marcela V. Maus, Umar Mahmood, Pedram Heidari

**Affiliations:** 1Division of Nuclear Medicine and Molecular Imaging, Department of Radiology, Massachusetts General Hospital, Boston, MA 02114, USA; 2Cellular Immunotherapy Program, Cancer Center, Massachusetts General Hospital, Harvard Medical School, Boston, MA 02114, USA

**Keywords:** CAR T-Cell therapy, melanoma, granzyme B PET imaging, NSG mice, humanized mouse model

## Abstract

**Background/Objectives**: Granzyme B (GZB) PET Imaging is a non-invasive tool that can determine tumoral and systemic effects in immunotherapy. We aim to evaluate ^68^Ga-NOTA-CYT-200 PET Imaging as a molecular imaging approach to determine CAR T-cell therapy response in a human melanoma mouse model. Our goal is to provide a method to monitor CAR T-cell therapy for patients with melanoma and other solid tumors. **Methods**: A human melanoma mouse model was generated by implanting naïve NSG mice (n = 28) with a human melanoma cell line (A375) subcutaneously (s.c.). After tumor implantation, mice were randomly assigned to receive either the treatment (CAR T) or vehicle solution (controls). After treatment, tumor sizes were measured every other day up to 35 days after cell implantation. ^68^Ga-NOTA-CYT-200 PET Imaging was performed on days 2, 7, and 14 after CAR T-cell administration to assess T-cell activity within the tumors and organs. The PET Imaging results were correlated with IHC and immunofluorescent staining and cytokine assessment of tumor samples. **Results**: Tracer uptake within tumors of the CAR T group was significantly greater on days 2 (3.1 ± 1.2 vs. 1.1 ± 0.4, *p* = 0.002) and 7 (2.0 ± 1.1 vs. 1.1 ± 0.1, *p* = 0.01) after treatment, even before the CAR T group first presented with significantly lower tumor volumes on day 11 after treatment (61.8 mm^3^ ± 8.7 vs. 287.1 mm^3^ ± 157.6, *p* = 0.05). GZB (*p* = 0.03) and CAR T (*p* = 0.001) staining were also significantly greater in tumors of CAR T-cell-treated mice. Inflammatory cytokines such as IFN gamma (*p* = 0.03), CXCL10 (*p* = 0.004), and CCL5 (*p* = 0.02) concentrations were also significantly greater in CAR T-cell-treated tumors. **Conclusions**: CAR-T-treated tumors show significantly elevated ^68^Ga-NOTA-CYT-200 uptake compared with controls, consistent with enhanced effector activity.

## 1. Introduction

Malignant melanoma is amongst the most diagnosed cancers in the United States, with an estimated 106,000 new cases in 2021 [1,2]. While its incidence increased over the past decade, particularly in developed, faired-skin countries, mortality rates decreased through overall improvement in available treatments, especially the development of targeted therapies in recent years [3,4]. Chimeric antigen receptor (CAR) T-cell therapy is a novel, cell-based immunotherapy approach that can be an alternative treatment for advanced melanomas that may have become resistant to chemotherapy or radiotherapy [5]. Despite huge pre-clinical and clinical efforts to translate CAR T-cell therapy into the clinical setting for melanoma patients, the process has been hindered by several challenges encountered during clinical trials, such as low response rates and serious side effects, ultimately resulting in delay or termination of many CAR T-cell therapy trials [6,7]. As reviewed by Sterner et al., clinical translation of CAR T-cell therapy used for the treatment of solid tumors continuously faces challenges, including low response rates and serious side effects [8]. Importantly, a significant proportion of patients fail to respond due to both under- and overtreatment mechanisms: undertreatment may result from insufficient CAR T-cell persistence, limited tumor infiltration, or early exhaustion of transferred T cells, while overtreatment can cause severe toxicities such as cytokine release syndrome or neurotoxicity that limit therapeutic efficacy and patient benefit [8,9,10,11]. Therefore, there is an urgent need for a reliable tool that can accurately monitor CAR T-cell treatment to ultimately enable further optimization and to significantly improve cell-based cancer treatments for patients with tumors like advanced melanoma.

Various biomarkers (PD-1, PD-L1, CD8, and CD3) are utilized to assess therapeutic efficacy and early treatment response in immunotherapies, which are mostly determined through biopsies, genetic screening, or molecular imaging [12,13,14,15], such as positron emission tomography (PET). PET is a non-invasive functional imaging modality that uses radiolabeled tracers to visualize and quantify physiological and molecular processes in vivo. It relies on the detection of pairs of gamma photons emitted indirectly by a positron-emitting radionuclide incorporated into biologically active molecules. These radiotracers distribute according to their biochemical properties, enabling the visualization of regional metabolic activity, receptor expression, or enzyme activity with high sensitivity and spatial resolution. PET Imaging is used in oncology as it allows for real-time, quantitative assessment of tumor microenvironment (TME) and immune cell function, aiding the evaluation of therapeutic response and disease progression. However, anatomic imaging (CT, MRI) can be associated with a tumoral immune cell infiltrate that can make responding tumors appear to grow, also referred to as pseudo-progression, thereby potentially decreasing its usefulness for early response assessment [16].

Granzyme B (GZB) is a key enzyme produced by cytotoxic T cells and natural killer cells that induces programmed cell death (apoptosis) in target cells. It does this by activating proteins inside the target cell that trigger the cell’s self-destruction. GZB is crucial for killing infected or cancerous cells and plays roles in inflammation and tissue remodeling. The prior literature supports the use of GZB-targeted PET Imaging as a generalized biomarker of cytotoxic T-cell activation across various immunotherapies [13,17]. We have previously shown that ^68^Ga-NOTA-CYT-200 PET Imaging can provide an early assessment of treatment response in checkpoint inhibitor therapy [18]. While immune checkpoint inhibition and CAR T-cell therapy represent mechanistically distinct immuno-therapeutic strategies, checkpoint blockade activates endogenous T cells, whereas CAR T-cell therapy involves adoptively transferred, engineered lymphocytes with different expansion kinetics and trafficking profiles. Our rationale for applying ^68^Ga-NOTA-CYT-200 PET Imaging in the response assessment of CAR T-cell therapy is grounded in targeting a convergent endpoint of antitumor immune response: the release of cytotoxic granules, especially GZB, by activated effector lymphocytes. Based on the results of previous studies [13,18], we hypothesize that ^68^Ga-NOTA-CYT-200 signal reflects the presence of the activity of cytotoxic effector molecules, particularly GZB, which is a shared terminal effector of immune-mediated tumor killing across various immune treatments.

## 2. Materials and Methods

### 2.1. Animal Studies

The animal study protocol received approval in accordance with guidelines from the Institutional Animal Care and Use Committee (2017N000163, approved on 8 October 2022). A total of thirty-eight male 8-week-old naïve NOD.Cg-Prkdc^scid^ Il2rg^tm1Wjl^/SzJ (NSG) mice were procured from the Jackson Laboratory (Bar Harbor, ME, USA). The mice were maintained in a Biosafety Level 2 facility under a 12 h light/dark cycle, with unrestricted access to a standard rodent chow diet (comprising approximately 18% protein, 5% fat, and 5% fiber) and water. At the time of tumor implantation (designated as day 0), the average body weight was 27.5 g ± 2.1. A decrease in body weight exceeding 15% from tumor implantation was established as a humane endpoint. Prior to experimentation, all animals were acclimated for a period of two weeks.

### 2.2. Cell Culture and Development of Human Melanoma Mouse Model

As pictured in Figure 1, human melanoma A375 cells (ATCC^®^) were maintained in Dulbecco’s Modified Eagle’s Medium (Thermo Fisher Scientific, Waltham, MA, USA) supplemented with 10% fetal bovine serum and 1% penicillin–streptomycin (Thermo Fisher Scientific) under standard culture conditions at 37 °C with 5% CO_2_. Twenty-eight male NSG mice received subcutaneous injections of 2 × 10^6^ A375 cells suspended in a 1:1 mixture of Matrigel (Corning™ Matrigel™ Membrane Matrix) and phosphate-buffered saline (Sigma-Aldrich, St. Louis, MO, USA) totaling 100 μL, delivered into the left shoulder region.

### 2.3. CAR T-Cell Treatment

Human-derived CAR T cells targeting the epidermal-growth factor receptor (EGFR) were stored in vials in liquid nitrogen prior to use. A375-bearing NSG mice were randomly assigned to receive intravenous (i.v.) injections of either 2 × 10^6^ CAR T cells (n = 15) suspended in 100 µL of RPMI medium (ATCC^®^) or vehicle solution as controls (n = 13). CAR T cells were administered four days after tumor implantation to allow for initial tumor engraftment and establishment of a measurable tumor burden, which is a standard approach in xenograft models to ensure reliable assessment of therapy response. This timing also aligns with the kinetics of CAR T-cell expansion and activation, maximizing the likelihood of observing therapeutic effects. Tumor size was assessed every other day for up to 35 days following tumor implantation using a digital caliper with a measurement accuracy of ±0.1 mm. Tumor volume was measured using calipers and calculated as
$V=\frac{1}{2}\times length\times {width}^{2}$.

### 2.4. ^68^Ga-NOTA-CYT-200 Labeling and PET Imaging

^68^Gallium was obtained by eluting a ^68^Ga/^68^Ge generator (Eckert and Ziegler, Berlin, Germany) with 0.1 M of hydrochloric acid. A total of 2 mL of the eluted ^68^Ga solution was combined with 100 μL of NOTA-CYT-200 (CytoSite Biopharma, Sudbury, MA, USA), prepared as a 1 mg/mL stock in metal-free water (OmniTrace Ultra, Millipore, Burlington, MA, USA) and buffered with 2 M of HEPES to adjust the pH to between 3.5 and 4.0. The reaction mixture was heated at 95 °C for 10 min and then neutralized to pH 7.0 using sodium hydroxide before intravenous administration via the tail vein. The radiochemical purity of the final preparation exceeded 95%.

A subset of mice (n = 9 CAR T; n = 8 controls) underwent PET Imaging 2 (PET D2), 7 (PET D7), and 14 (PET D14) days after CAR T-cell administration to capture the dynamic changes in tracer uptake and tumor response. A subset of mice (n = 9 CAR T-treated; n = 8 controls) underwent PET Imaging at 2, 7, and 14 days post-CAR T-cell treatment to monitor tracer uptake and tumor response dynamics. Each mouse received an injection of 4.8 ± 0.4 MBq ^68^Ga-NOTA-CYT-200 60 ± 5 min prior to PET/CT scanning. Images were acquired using a small-animal Argus PET/CT system (Sedecal, Madrid, Spain) in static mode over two bed positions for 20 min. PET reconstruction employed 2D-OSEM with 2 iterations and 16 subsets and was corrected for random and scatter events. Two independent blinded reviewers analyzed the images using VivoQuant™ software, version 4.0 (InVicro). Standardized uptake values (SUVs) for the tumor, liver, lungs, and colon were normalized to the blood pool activity, which was defined by drawing a region of interest (ROI) around the left ventricle of the heart. This normalization generated target-to-background ratios—tumor-to-blood (TBR), liver-to-blood (LBR), lung-to-blood (LuBR), and colon-to-blood (CBR)—accounting for nonspecific background signals and enabling comparison across different groups and imaging timepoints.

### 2.5. Immunohistochemical (IHC) and Immunofluorescent Staining

Tumor tissue samples were dissected two days after treatment for ex vivo correlation of timepoint PET D2. Immunohistochemical staining for CD3 (ab135372; Abcam, Cabridge, UK) and CD8 (ab217344, Abcam), as well as immunofluorescent staining for GZB and mCherry, was performed on paraffin-embedded tumor samples (n = 5/group). Staining was conducted using a rabbit anti-human granzyme B primary antibody (ab243879; Abcam), followed by detection with an Alexa Fluor Plus 647 highly cross-adsorbed goat anti-rabbit IgG secondary antibody (a32733; Thermo Fisher) [13]. Microscopy images were acquired through the Brigham Women’s Hospital Pathology Core Services, and quantitative analysis was performed using ImageJ2 software (Version 2.9.0; Fiji).

### 2.6. Cytokine Analysis

Tumors were extracted two days after treatment for cytokine analysis using previously described methods [19]. Briefly, tumor samples were cultured at 37 °C and 5% CO_2_ for 24 h in 1.5 mL of RPMI medium (ATCC^®^) and stored at −80 °C until further processing. Supernatant was analyzed using a Cytokine & Chemokine 34-Plex Human ProcartaPlex™ Panel 1A (Thermo-Fisher) for detection and quantification of targets. A total of 50 µL of 1× diluted magnetic beads was added to each well of the 96-well plate. Following definition of the plate map, serial diluted antigen standard and tumor samples of the CAR T (n = 5) and control (n = 5) group were added to the wells and incubated at room temperature shaking (500 rpm) for 60 min. In total, 25 µL of Detection Antibody Mixture (1X) was added to each well and incubated for another 60 min under the same conditions followed by 30 min of incubation in Streptavidin R-Phycoerythrin Conjugate (SAPE) solution. Magnetic beads were resuspended in 120 µL of reading buffer mixed with samples and read as duplicates on a Luminex^®^-200 Instrument. The concentrations of the samples were calculated by plotting the expected concentration of the standards against the net mean fluorescence imaging (MFI) generated by each standard. Datasets were exported from the ProcartaPlex™ Analysis App on Thermo Fisher Connect.

### 2.7. Statistical Analyses

Quantitative data analysis was conducted using Prism software (version 10; GraphPad). A power calculation, based on expected effect size and variability from preliminary results, was performed to determine the necessary sample size. The relatively small sample size was justified by the reproducibility of the xenograft model and the statistical significance observed in the primary outcomes. Data are presented as mean ± SEM. Comparisons of means between two groups were performed using unpaired Student’s *t*-tests as appropriate. Differences among multiple treatment groups were assessed using one- or two-way ANOVA followed by Tukey’s multiple comparison test. A *p*-value of less than 0.05 was considered statistically significant.

## 3. Results

### 3.1. CAR T-Cell Therapy Delays Tumor Growth of Melanoma Tumors

Tumor growth measurements (Figure 2A) demonstrated that CAR T-cell therapy administered 4 days after tumor implantation leads to a tumor growth delay (CAR T: day 4: 208.0 mm^3^ ± 80.1 vs. day 35: 263.6 mm^3^ ± 192.4, *p* = 0.95; control: day 4: 190.9 mm^3^ ± 84.4 vs. day 35: 972.7 mm^3^ ± 198.5, *p* < 0.0001), and significantly decreases the tumor volumes of the CAR T group compared with the control group as early as eleven days after tumor implantation (61.8 mm^3^ ± 8.7 vs. 287.1 mm^3^ ± 157.6, *p* = 0.05) up to the final measurement timepoint, thirty-five days after tumor implantation (263.6 mm^3^ ± 192.4 vs. 972.7 mm^3^ ± 198.5, *p* < 0.0001).

Body weight measurements (Figure 2B) found a similar slight reduction in weight (in grams) for both groups after tumor implantation and treatment administration. However, the body weight for both treatment groups was similar on day 2 compared with day 35 after tumor implantation (CAR T: 28.6 g ± 2.2 vs. 26.5 g ± 2.0, *p* = 0.97; controls: 29.0 g ± 3.8 vs. 29.0 g ± 0.2, *p* > 0.9999), and there was no difference at the final timepoint between the two groups (CAR T: 26.6 g ± 2.0 vs. controls: 29.0 g ± 0.2, *p* = 0.99).

### 3.2. ^68^Ga-NOTA-CYT-200 PET Imaging Predicts Treatment Response of Tumors Treated with CAR T Cells

^68^Ga-NOTA-CYT-200 PET Imaging timepoints PET D2 (3.1 ± 1.2 vs. 1.1 ± 0.4, *p* = 0.001) and PET D7 (2.0 ± 1.1 vs. 1.1 ± 0.1, *p* = 0.01) found a significantly greater tracer uptake within tumors of the CAR T group, depicted as the tumor-to-blood ratio (TBR) (Figure 3B). The greatest TBR was observed on PET D2, five days before a significant difference in tumor volumes was detected (Figure 3A), and seven days following i.v. CAR T-cell administration (61.8 mm^3^ ± 8.7 vs. 287.1 ± 157.6, *p* = 0.05). On PET D14, tracer uptake was reduced in the CAR T group compared to PET D2 (3.1 ± 1.2 vs. 1.9 ± 0.7, *p* = 0.06), with similar uptake compared to the controls on PET D14 (1.9 ± 0.7 vs. 1.0 ± 0.4, *p* = 0.53). This reduction in TBR on PET D14 was subsequently followed by an increase in tumor volume up to the final measurement timepoint (24.4 mm^3^ ± 9.5 vs. 263.6 mm^3^ ± 192.4, *p* = 0.2).

### 3.3. CAR T-Cell-Treated Mice Showed Greater Tracer Uptake in the Colon

^68^Ga-NOTA-CYT-200 uptake within similar regions of interest (ROIs) within the colon, liver, and lung for both groups was normalized against the blood pool measured in the heart region and is presented as the colon-to-blood ratio (CBR), liver-to-blood ratio (LBR), and lung-to-blood ratio (LuBR) (Figure 4A–C, respectively). We found a greater CBR for the CAR T group on PET D2 (8.3 ± 4.7 vs. 2.9 ± 2.9, *p* = 0.08) and PET D7 (5.4 ± 3.7 vs. 1.6 ± 1.1, *p* = 0.47). CBR was similar in both groups on PET D14 (3.9 ± 2.0 vs. 1.8 ± 0.7, *p* = 0.82), indicating a decrease in tracer uptake over time. The CBR of the control group was similar on PET D2 and PET D7 (2.9 ± 2.9 vs. 1.6 ± 1.1, *p* = 0.86) and between PET D7 and PET D14 (1.6 ± 1.1 vs. 1.8 ± 0.7, *p* = 0.99). LBR was similar at all imaging timepoints for both treatment groups, although there was a slight difference on PET D2 compared to PET D14 in the CAR T group (7.2 ± 2.3 vs. 3.7 ± 0.8, *p* = 0.06). However, there was no significant difference in LBR between the groups at any of the three imaging timepoints (D2: *p* = 0.57, D7: *p* > 0.9999, D14: *p* = 0.99). LuBR of the CAR T group was similar to that of the controls on PET D2 (1.7 ± 0.8 vs. 1.6 ± 1.1, *p* = 0.99), PET D7 (1.2 ± 0.2 vs. 1.2 ± 0.1, *p* > 0.9999), and PET D14 (1.1 ± 0.4 vs. 1.2 ± 0.5, *p* = 0.97).

**Figure 4 diagnostics-15-03058-f004:**
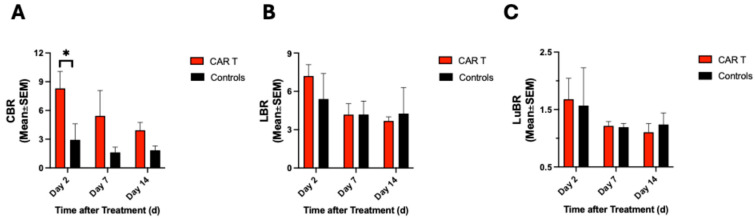
^68^Ga-NOTA-CYT-200 PET Imaging found greater CBR (**A**) in the CAR T group (n = 9) compared to controls (n = 8) two days after treatment (*p* = 0.08). There is a trend for greater CBR in the CAR T group on days 7 and 14 after treatment, with a continuous decrease over time; however, these differences were not significant (*p* = 0.47 and *p* = 0.82, respectively). There was no significant difference in LBR (**B**) and LuBR (**C**) between the group receiving CAR T treatment and the control group. * *p* ≤ 0.05 is considered significant. Two-way ANOVA with multiple comparisons. 95% CI CBR: 0.65–6.87%, LBR: −1.78–2.59%, LuBR: −0.5–0.49%.

### 3.4. Tumor Samples Were Analyzed by Immunostaining (IHC and Immunofluorescence) to Assess Granzyme B Expression and CAR T-Cell Infiltration

The presence of GZB and CAR-T cells in tumor tissue was confirmed by IHC and immunofluorescent staining. Staining of extracted tumor tissue samples (Figure 5A,C) for GZB showed greater expression levels of the CAR T group compared with the controls (16.17 ± 10.75 vs. 3.26 ± 1.5; *p* = 0.03; Figure 5B). Staining for mCherry/CAR T also found significantly greater staining within tumors treated with CAR T-cell therapy compared with controls (13.57 ± 3.55 vs. 4.09 ± 2.63; *p* = 0.001; Figure 5D). With staining for CD3 and CD8, we were able to detect these immune cell populations within the extracted tumors of treated mice (see Appendix A).

### 3.5. CAR T-Cell Therapy Is Associated with a Targeted Increase in Inflammatory Mediators, Consistent with Activation of an Antitumor Immune Response

Cytokine profiling of extracted tumor samples was performed using a multiplex bead-based immunoassay (Thermo Fisher Scientific) to assess local immune responses following treatment. Among the panel of cytokines analyzed, a subset of cytokines demonstrated significantly elevated concentrations in the CAR T group. Specifically, IL-6 and IL-18 levels were increased compared with controls (92.91 ± 52.93 vs. 13.78 ± 6.16, *p* = 0.01, and 31.58 ± 10.03 vs. 18.21 ± 5.75, *p* = 0.04, respectively). Further, IFN-γ was elevated in the CAR T group (39.74 ± 33.28 vs. 0.95 ± 0.39, *p* = 0.03), as well as CCL3 (41.39 ± 16.35 vs. 6.85 ± 1.48, *p* = 0.002), CCL4 (126.03 ± 60.06 vs. 40.43 ± 9.08, *p* = 0.02), and CCL5 (13.09 ± 6.79 vs. 1.55 ± 0.16, *p* = 0.02), and IP-10 (CXCL10) was also significantly higher in the CAR T group (543.36 ± 290.94 vs. 15.68 ± 3.83, *p* = 0.004). CCL2 (687.41 ± 401.55 vs. 262.64 ± 95.05) was also increased in the CAR T group compared with the controls; however, this difference was not statistically significant (*p* = 0.05). A heat map visualization summarizing the cytokine expression profiles for both groups is presented in Figure 6, illustrating the selective upregulation of proinflammatory cytokines in the CAR T group. The precise quantitative comparisons for each cytokine are presented in the Appendix A, illustrating the significant elevation of these biomarkers following CAR T-cell therapy in direct comparison between the two groups.

**Figure 6 diagnostics-15-03058-f006:**
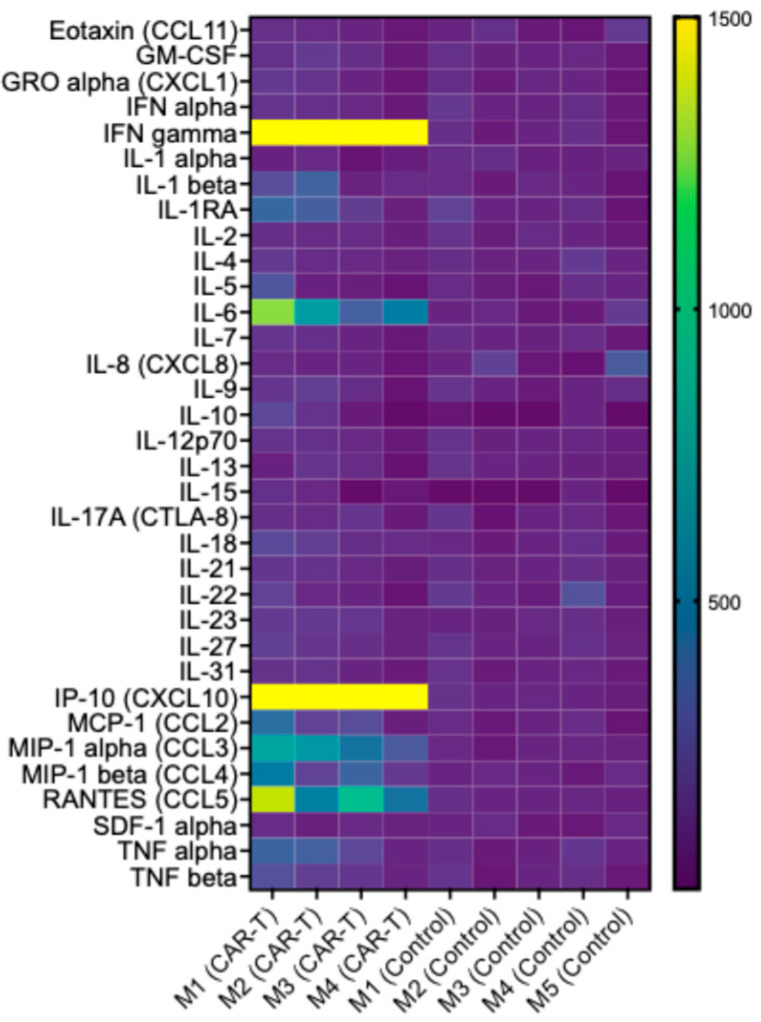
Heat map representation of cytokine expression profiles in tumor samples from the CAR T and control groups. IL-6 (*p* = 0.01), IL-18 (*p* = 0.04), IFN-γ (*p* = 0.03), CCL3 (*p* = 0.002), CCL4 (*p* = 0.02), CCL5 (*p* = 0.02), and IP-10/CXCL10 (*p* = 0.004) were significantly elevated in the CAR T group compared with controls. CCL2 levels were also higher in the CAR T group but did not reach statistical significance (*p* = 0.05). Each column represents an individual tumor sample, and each row corresponds to a measured cytokine. Color intensity reflects relative cytokine concentration, with red indicating higher and blue indicating lower expression levels.

## 4. Discussion

This study is the first to evaluate the efficacy of GZB PET Imaging in detecting antitumor T-cell activity related to CAR T-cell therapy in a human melanoma mouse model. While multiple tools exist for detecting treatment response in CAR T-cell therapy, its major disadvantage is complexity and difficulty in design and operation [20,21,22,23,24]. Radionuclide-based T-cell imaging techniques or reporter gene strategies have shown their ability to visualize CAR T cells in vivo, thereby providing crucial information about cell viability and biodistribution. However, these techniques require in vitro labeling of the CAR T cells with either radionuclides or radioactive materials prior to infusion or usage of reporter gene-transduced T cells expressing an enzyme, receptor, or transporter. Molecular imaging, and more particularly ^68^Ga-NOTA-CYT-200 PET Imaging, can be used as an early tool to permit quantitative measurement of the tumor activity [18], thereby providing an indirect method for determining CAR T-cell treatment response without the need for pre-infusion manipulation or direct radiolabeling of the CAR T cells. ^68^Ga-NOTA-CYT-200 is a next-generation PET tracer developed to bind the active form of human GZB. It is composed of a GZB-specific peptide coupled to a NOTA chelator, enabling efficient radiolabeling with gallium-68 for PET Imaging. NOTA-CYT-200 exhibits a nanomolar inhibition constant (Ki 4.2 nM), reflecting strong and selective affinity for active GZB while minimizing off-target interactions. Its molecular design promotes rapid clearance from non-target tissues, which enhances tumor-to-background contrast and overall imaging precision. Compared with earlier GZB-directed tracers, NOTA-CYT-200 was chosen for its superior specificity, improved pharmacokinetic profile, and compatibility with humanized systems, making it a robust tool for visualizing functional cytotoxic T-cell activity during the effector phase. A major benefit of this tracer—central to this study—is its ability to reveal active immune responses at an early stage, potentially before measurable tumor changes occur [18]. In this study, we demonstrated that ^68^Ga-NOTA-CYT-200 PET Imaging can quantify CAR T cells and T-cell subsets through GZB uptake in melanoma tumors treated with CAR T-cell therapy even before measurable tumor size regression. ^68^Ga-NOTA-CYT-200 PET Imaging showed that administration of 2 × 10^6^ CAR T cells leads to significantly greater GZB uptake within treated tumors as early as two days after cells were administered, while measurable tumor size reduction could only be observed between four and six days later. Overall, we observed a significant tumor growth delay over 31 days after the treatment was administered. Interestingly, tracer uptake within the treated tumors was lower on PET D7 and even more on PET D14. While CAR T-cell therapy has significantly improved the outcome for hematologic and lymphatic tumors, it has not yet been successfully translated into the clinic for the treatment of solid tumors, including melanomas, for reasons that include different interaction pathways and the immunosuppressive TME [25,26], which ultimately leads to CAR T-cell exhaustion, making TME properties one of the greatest challenges for CAR T-cell treatment efficacy in solid tumors [26,27]. Prolonged activation and exposure to tumor antigens have previously been described to reduce the proliferation and expansion of CAR T cells and to impair cytokine secretion and cytotoxicity, altogether leading to the exhaustion of CAR T cells [28]. The decline in tracer uptake at later timepoints, accompanied by subsequent tumor regrowth, is consistent with—but does not definitely prove—a process of CAR T-cell functional decline and/or exhaustion. While the pattern may be compatible with exhaustion or waning effector function in the context of an immunosuppressive TME, alternative explanations such as altered trafficking or antigen modulation cannot be excluded.

Interestingly, we observed transiently increased tracer uptake in the intestinal region of mice that received CAR T-cell therapy. However, as the EGFR-targeted CAR employed in this study does not cross-react with murine EGFR, the observed colon uptake is unlikely to represent on-target binding to endogenous EGFR. Given this lack of target interaction, we hypothesize that the increased tracer uptake in the colon reflects off-target or nonspecific immune activation rather than direct EGFR-mediated effects. This could be due to cytokine-driven inflammation, secondary immune effects associated with CAR T-cell therapy, or local immune cell infiltration unrelated to EGFR specificity. Based on current evidence and the specificity of the CAR construct used, we conclude that colon uptake is unlikely to be an on-target effect and instead reflects a broader immune response following CAR T-cell treatment. This finding is more plausibly attributed to nonspecific immune activation or immune-related inflammatory responses induced by treatment, as previously demonstrated by increased intestinal tracer accumulation in models of immune-mediated colitis and systemic T-cell activation. Similar findings have been reported for ^68^Ga-NOTA-CYT-200 and other granzyme B-based PET Imaging in immunotherapy contexts, where intestinal signal elevation correlates with immune-related inflammation rather than target expression [12,18,29]. However, clarifying this in future experiments would benefit from direct immune cell marker analysis and tissue cytokine assay. Therefore, the mechanistic basis of ^68^Ga-NOTA-CYT-200 uptake in the colon remains uncertain and should be interpreted as an exploratory finding that requires validation in future studies.

There are some limitations that need to be considered for this study. First, while ^68^Ga-NOTA-CYT-200 uptake corresponds to cytotoxic lymphocyte activity, this study cannot directly attribute the PET signal exclusively to transferred CAR T cells in vivo, as other immune cell subsets may contribute to tracer accumulation in the TME. Second, the observation of increased tracer uptake in the colon cannot be fully explained, because no colon histology, colon-specific cytokine measurements, or dedicated immune cell staining of colonic tissue were performed in this model. Third, the sample sizes of the PET-imaged cohort and of the cytokine-analysis subgroup were relatively small, which may limit the generalizability and statistical power of some of the observed effects. The relatively small sample size was justified by the reproducibility of the xenograft model and the statistical significance overserved in the primary outcomes, although this remains a limitation and may affect the generalizability of the secondary exploratory endpoints. Finally, these findings are derived from a single human melanoma xenograft model in immunodeficient mice and may not fully capture the complexity of human solid tumors and their immune microenvironment.

Despite these limitations, we were able to demonstrate greater GZB expression through IHC and immunofluorescent staining in tumors of mice that received CAR T-cell therapy. Overall, these findings support that the antitumor effect, such as the greater GZB signal observed in vivo and ex vivo, could correlate with the cytotoxic activity of the infused CAR T cells. We then performed an assessment of cytokine profiles on tumor samples and found greater expression levels of certain cytokines (IFN gamma) and chemokines (IP-10, MCP-1, MIP-1 beta, and RANTES) in CAR T-cell-treated tumors. These findings suggest that tumoral T-cell infiltration is related to CAR T-cell therapy. IFN gamma is a cytokine that is produced by T cells, including CAR T cells, with multiple antitumor effects such as enhanced cytotoxicity, promotion of antigen presentation, and immune cell activation. Larson et al. demonstrated the important role of IFN-gamma specifically produced by CAR T cells in tumor cell killing, while revealing that the interferon gamma receptor signaling pathway is critical for the cytotoxic susceptibility of glioblastoma tumors to CAR T-cell therapy [25]. RANTES (CCL5) has been shown to modulate cytokine production, inducing a switch from Th2-type to Th1-type cytokines, and the amount of CCL5 expression at the tumor site determines the effectiveness of the antitumor response, which is associated with the infiltration of an increased number of NK, CD4, and CD8 cells at the tumor site [30]. IP-10 (CXCL10), which is a chemokine involved in the recruitment and activation of immune cells, also plays a role in CAR T-cell therapy by attracting T cells, NK cells, and other immune cells to the tumor site, and it enhances the infiltration of CAR T cells into the tumor site, thereby promoting their antitumor activity. Studies have explored various strategies to enhance the production or secretion of RANTES, including gene modifications in CAR T cells to overexpress RANTES or engineering the tumor cells to produce RANTES. These approaches aim to improve CAR T-cell infiltration into solid tumors and enhance their antitumor activity. Further, we detected greater levels of MCP-1 (CCL2) in the CAR T-cell-treated tumors, which is a chemokine involved in recruiting monocytes and macrophages to inflammatory sites. Increased expression of CCL2 has been reported in the TME of patients undergoing CAR T-cell therapy. Overall, CAR T-cell therapy does not merely introduce new effector cells but invigorates and reshapes the pre-existing immune microenvironment, catalyzing inflammatory mediator secretion, T-cell activation, and recruitment [25,30]. It needs to be considered that the response prediction of CAR T-cell therapy depends on non-tumor-specific factors such as the TME, but also CAR T-cell subset composition, functional state, and activation patterns of infused CAR T cells, altogether impeding the identification of patients more likely to respond to CAR T-cell therapy [31].

## 5. Conclusions

In conclusion, CAR-T-treated tumors exhibited significantly higher uptake of ^68^Ga-NOTA-CYT-200 compared to controls, indicating increased effector immune activity within the tumor microenvironment. This finding is supported by ex vivo analyses showing elevated granzyme B levels and infiltration of cytotoxic lymphocytes within the tumor, indicating that tracer uptake correlates with granzyme B activity and CAR T-cell infiltration consistent with active immune-mediated tumor cell killing.

## Figures and Tables

**Figure 1 diagnostics-15-03058-f001:**
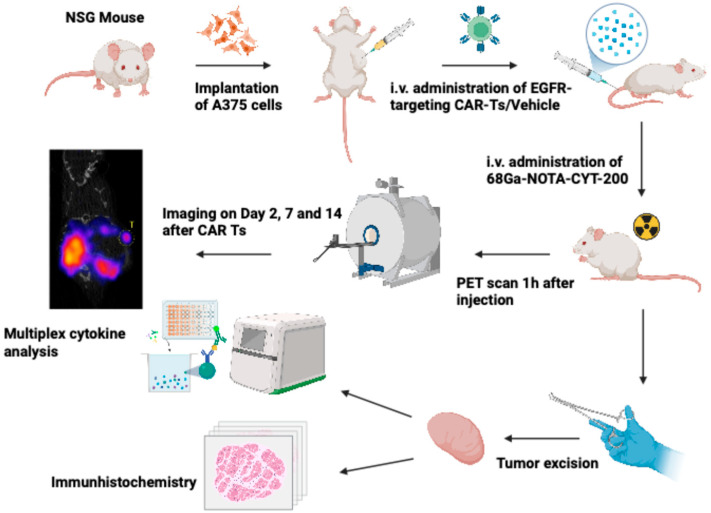
Graphical overview of the experimental design. Created in BioRender. Summer, P. (2025).

**Figure 2 diagnostics-15-03058-f002:**
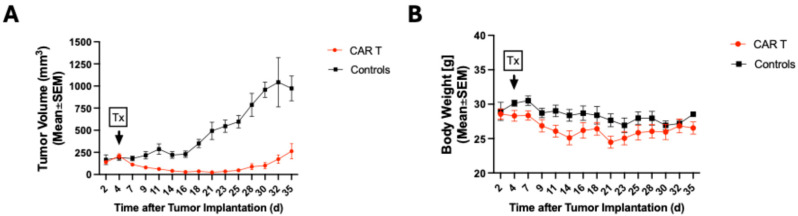
(**A**) Tumor growth measurements were performed from two to thirty-five days after tumor implantation (CAR T: n = 9; control: n = 8) to monitor treatment response. The CAR T group presented with a tumor growth delay, with tumor volumes increasing above pre-treatment volumes 31 days after CAR T-cell administration. (**B**) There was a similar average body weight (in grams) for both groups without a significant reduction throughout the experiment.

**Figure 3 diagnostics-15-03058-f003:**
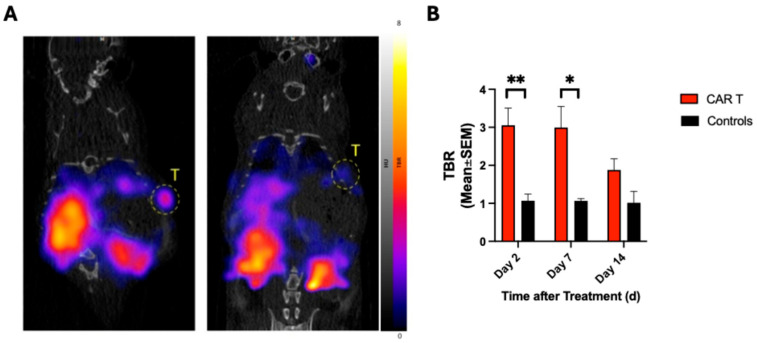
(**A**,**B**) ^68^Ga-NOTA-CYT-200 PET images on PET D2 demonstrate significantly greater TBR within tumors of the group that received CAR T-cell therapy (left) compared to the control group (right), representing greater tracer uptake of treated tumors. T = tumor, L = liver, K = kidney. ** *p* ≤ 0.01, * *p* ≤ 0.05. Two-way ANOVA with multiple comparisons. 95% CI: 0.88–2.3%.

**Figure 5 diagnostics-15-03058-f005:**
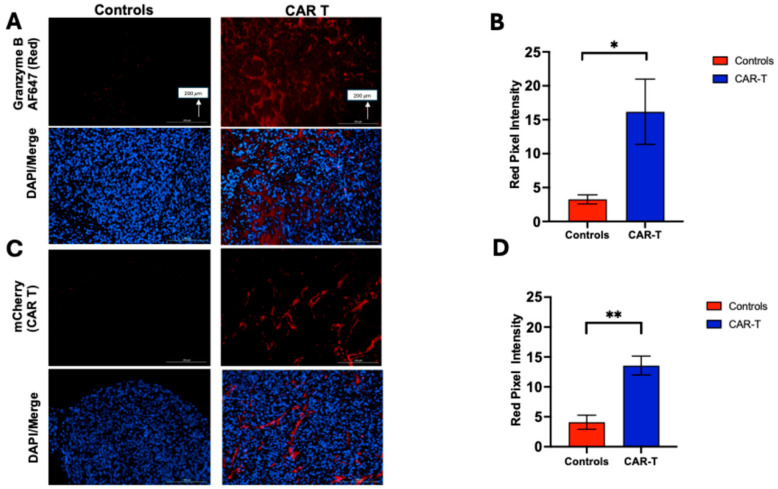
Representative immunofluorescent images of tumor tissue samples stained for GZB (**A**), and mCherry/CAR T cells (**C**) compared between the untreated (**left**) and treated (**right**) groups. The difference was significant for GZB (**B**) and mCherry (**D**) (** *p* ≤ 0.01, * *p* ≤ 0.05). Unpaired *t*-test. Scale bar represents 200 μm (µm).

## Data Availability

Data presented in this study are contained within this article.

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
