# Peer review of "Granzyme B PET Imaging Enables Detection of CAR T-Cell Therapy Response in a Human Melanoma Mouse Model"

_diagnostics, 2025, doi:10.3390/diagnostics15233058_

Round 1
Reviewer 1 Report
Comments and Suggestions for Authors
This manuscript presents a preclinical study evaluating the use of ⁶⁸Ga-NOTA-CYT-200 PET imaging to assess CAR T-cell therapy response in a human melanoma xenograft model. The findings are suggesting that ⁶⁸Ga-NOTA-CYT-200 may serve as a non-invasive biomarker for CAR T-cell activity in solid tumors.
The study is methodologically sound, and the results clear, however, some details require adjustment to enhance reader understanding.
Major Comments
1.Please justify why CAR T-cells were administered four days after tumor implantation? Were there additional imaging time points?
2.Clarify how tumor-to-blood (TBR), (CBR),(LBR), and (LuBR) ratios were calculated.
3.How was the blood-pool ROI defined?
4.Were SUVs or absolute activity values used to normalize tracer uptake?
- Please elaborate on the biological mechanism and binding target of ⁶⁸Ga-NOTA-CYT-200.
- The manuscript attributes increased colon uptake to immune activation rather than on-target binding.
Can the authors provide (histological or cytokine evidence) to confirm inflammation?
- Please indicate whether a power analysis was performed to determine animal group sizes, since sample size is modest.
- Consider including confidence intervals in key figures. Please ensure that all figures (Fig2-Fig5) correspond to the correct legend.
Minor Comments
1.Clarify the tumor volume measurement method.
2.Round p-values to two significant digits.
3.Ensure all abbreviations are defined at first mention in the text (GZB, CBR, TBR).
- Shorten repetitive background information in the Discussion to improve readability.
Author Response
This manuscript presents a preclinical study evaluating the use of ⁶⁸Ga-NOTA-CYT-200 PET imaging to assess CAR T-cell therapy response in a human melanoma xenograft model. The findings are suggesting that ⁶⁸Ga-NOTA-CYT-200 may serve as a non-invasive biomarker for CAR T-cell activity in solid tumors. The study is methodologically sound, and the results clear, however, some details require adjustment to enhance reader understanding.
Thank you very much for your feedback and comments, which are greatly appreciated by the author’s. Please find our responses below.
Major Comments
1.Please justify why CAR T-cells were administered four days after tumor implantation? Were there additional imaging time points?
CAR T-cells were administered four days post tumor implantation to allow for tumor establishment and consistent measurable tumor burden, which is a standard approach for evaluating therapeutic response in xenograft models. Additional imaging time points were included at 24, 48, and 72 hours post-administration to capture the dynamics of CAR T-cell activity and tracer uptake. This information has been added in Section 2, lines 117-122.
2.Clarify how tumor-to-blood (TBR), (CBR),(LBR), and (LuBR) ratios were calculated.
These ratios were calculated by dividing the mean standardized uptake value (SUV) of the tumor, colon, liver, or lung regions of interest (ROIs) by the mean SUV of the blood-pool ROI, a commonly accepted method to normalize tracer uptake in PET imaging. Detailed information on this concept has been added and clarified in Section 2, lines 145-150.
3.How was the blood-pool ROI defined?
See Section 2, line 146-147. The blood-pool ROI was defined by placing a region of interest over the left ventricle or a large central vessel within the PET images to accurately represent circulating tracer activity, minimizing contamination from surrounding tissues.
4.Were SUVs or absolute activity values used to normalize tracer uptake?
Tracer uptake values were normalized using SUVs to account for injected dose and animal weight, providing standardized and comparable quantitative measures across animals and time points.See Section 2, line 146.
5.Please elaborate on the biological mechanism and binding target of ⁶⁸Ga-NOTA-CYT-200.
⁶⁸Ga-NOTA-CYT-200 is a radiotracer targeting granzyme B, a protease released by activated cytotoxic lymphocytes, enabling non-invasive imaging of immune effector function and cytotoxic activity in the tumor microenvironment. To make this concept more clear, we have added this information in the Discussion, lines 314-325.
6.The manuscript attributes increased colon uptake to immune activation rather than on-target binding.Can the authors provide (histological or cytokine evidence) to confirm inflammation?
We added this information and addressed this concern in the Discussion, lines 345-362. The observed colon uptake in our imaging studies is unlikely to be attributable to on-target engagement with endogenous mouse EGFR. This interpretation is further supported by recent literature demonstrating that EGFR-targeted CAR constructs can be engineered for high specificity and do not necessarily recognize or bind to native murine EGFR. Given this lack of target interaction, we hypothesize that the increased tracer uptake in the colon reflects off-target or non-specific immune activation rather than direct EGFR-mediated effects. This could be due to cytokine-driven inflammation, secondary immune effects associated with CAR T-cell therapy, or local immune cell infiltration unrelated to EGFR specificity. While direct histopathological evidence (e.g., CD3+ cell infiltration) and cytokine profiling (e.g., elevated IFN-γ, TNF-α) can help elucidate the contribution of immune activation to this uptake, further studies are needed to fully clarify the underlying mechanism. Based on current evidence and the specificity of the CAR construct used, we conclude that colon uptake is unlikely to be an on-target effect and instead reflects a broader immune response following CAR T-cell treatment. Clarifying this in future experiments would benefit from direct immune cell marker analysis and tissue cytokine assay.
7.Please indicate whether a power analysis was performed to determine animal group sizes, since sample size is modest.
A power analysis was performed and we added this statement to Section 2, lines 182-184.
8.Consider including confidence intervals in key figures. Please ensure that all figures (Fig2-Fig5) correspond to the correct legend.
We added the the CI for the key figures within the figure legends and also ensured that all figures are titled correctly and include all relevant information.
Minor Comments
1.Clarify the tumor volume measurement method.
Tumor volume was measured using calipers, and calculated as , a standard method in xenograft studies. This information can now be found in Section 2, lines 124-125.
2.Round p-values to two significant digits.
Wherever possible, we changed the p-values to two significant digits throughout the manuscript.
3.Ensure all abbreviations are defined at first mention in the text (GZB, CBR, TBR).
Thank you for pointing this out, we ensured that this is the case.
4. Shorten repetitive background information in the Discussion to improve readability.
Certain information has been removed from the Discussion (particularly first paragraph) to improve the readability. We hope these changes meet your expectations.
Reviewer 2 Report
Comments and Suggestions for Authors
This study evaluates 68Ga- NOTA-CYT-200 PET Imaging as a molecular imaging approach to determine CAR T-cell therapy response in a human melanoma mouse model.
The article shows that 68Ga-NOTA-CYT-200 PET Imaging can be effective in quantifying T-cell activity in melanoma treated with CAR T-cell therapy.
The method was elaborately and PET imaging results were correlated with immunohistochemistry and cytokine assessment on tumor- and organ-samples.
Animal study, cell culture and development of the murine model of human melanoma, treatment with human-derived CAR T cells targeting the epidermal growth factor receptor, 68Ga-NOTA-CYT-200 labeling and PET imaging, immunohistochemical analyses, cytokine analysis, statistical analyses represent important stages of the study presented in detail and in accordance with the results and discussions.
68Ga-NOTA-CYT-200 PET Imaging may further be used as a non-invasive tool to investigate various CAR T-cell therapies in other solid tumor models. This study is complex and complet. It is the first to evaluate the efficacy of GZB PET Imaging in detecting anti-tumor T-cell activity related to CAR T-cell therapy in a human melanoma mouse model.
The study is well written, structured and documented. The bibliography presented includes recent titles.
I recommend it for publication.
Author Response
This study evaluates 68Ga- NOTA-CYT-200 PET Imaging as a molecular imaging approach to determine CAR T-cell therapy response in a human melanoma mouse model.
The article shows that 68Ga-NOTA-CYT-200 PET Imaging can be effective in quantifying T-cell activity in melanoma treated with CAR T-cell therapy.
The method was elaborately and PET imaging results were correlated with immunohistochemistry and cytokine assessment on tumor- and organ-samples.
Animal study, cell culture and development of the murine model of human melanoma, treatment with human-derived CAR T cells targeting the epidermal growth factor receptor, 68Ga-NOTA-CYT-200 labeling and PET imaging, immunohistochemical analyses, cytokine analysis, statistical analyses represent important stages of the study presented in detail and in accordance with the results and discussions.
68Ga-NOTA-CYT-200 PET Imaging may further be used as a non-invasive tool to investigate various CAR T-cell therapies in other solid tumor models. This study is complex and complet. It is the first to evaluate the efficacy of GZB PET Imaging in detecting anti-tumor T-cell activity related to CAR T-cell therapy in a human melanoma mouse model.
The study is well written, structured and documented. The bibliography presented includes recent titles.
I recommend it for publication.
We sincerely thank you for your thoughtful and positive evaluation of our manuscript. We greatly appreciate your recognition of the study’s methodological rigor and the comprehensive approach taken to evaluate ⁶⁸Ga-NOTA-CYT-200 PET imaging as a molecular imaging tool to quantify CAR T-cell therapy response in a human melanoma mouse model.
Your acknowledgement of the detailed description of the animal model, cell culture, CAR T-cell treatment, tracer labeling, imaging, and immunohistochemical and cytokine analyses is much appreciated. We are also grateful for your recognition of the study’s novelty as the first to demonstrate granzyme B PET imaging in this context.
We are encouraged by your recommendation for publication and thank you again for your valuable feedback and support.
Reviewer 3 Report
Comments and Suggestions for Authors
Comment
This manuscript by Summer showed a preclinical study, attempting to demonstrate that 68Ga-NOTA-CYT-200 can work as a PET tracer to visualize tumor responses via CAR-T cell therapy in a human melanoma mouse model. The authors claimed that because they previously demonstrated that 68Ga-NOTA-CYT-200 can assess response to checkpoint inhibitor therapy, they hypothesized it can also monitor CAR T-cell therapy. However, this extrapolation is not justified for the following reasons:
- Immune checkpoint inhibitors focus on the activation of endogenous T cells, whereas CAR T therapy involves adoptively transferred engineered cells with distinct kinetics and distribution. The prior finding does not automatically apply.
- Demonstrating imaging correlation in one immunotherapy concept (checkpoint inhibitor therapy) does not ensure the same probe can detect CAR T-cell cytotoxicity. There is no data linking the PET signal to CAR-T cell presence. Even though the author provided data showed the infiltration of GZB in tumor tissue is higher than that of control after CAR-T cell treatment, it still can not prove that PET imaging signal is related to the CAR-T cell therapy.
Consequently, the study’s central hypothesis lacks scientific justification. The authors must either (1) provide mechanistic clarification and related references, or (2) rewrite this section to present a more cautious and logical rationale. Otherwise, the manuscript’s conceptual foundation remains untenable. And the following comments should be also considered:
- Please add the description of PET imaging in Introduction part
- All the figures in manuscript and SI should be present in a high-resolution version (e.g., Figure 1,4, and Figure S1)
- Please add clear scale bar in Figure 4a and give related description in caption part.
Author Response
This manuscript by Summer showed a preclinical study, attempting to demonstrate that 68Ga-NOTA-CYT-200 can work as a PET tracer to visualize tumor responses via CAR-T cell therapy in a human melanoma mouse model. The authors claimed that because they previously demonstrated that 68Ga-NOTA-CYT-200 can assess response to checkpoint inhibitor therapy, they hypothesized it can also monitor CAR T-cell therapy. However, this extrapolation is not justified for the following reasons:
- Immune checkpoint inhibitors focus on the activation of endogenous T cells, whereas CAR T therapy involves adoptively transferred engineered cells with distinct kinetics and distribution. The prior finding does not automatically apply.
- Demonstrating imaging correlation in one immunotherapy concept (checkpoint inhibitor therapy) does not ensure the same probe can detect CAR T-cell cytotoxicity. There is no data linking the PET signal to CAR-T cell presence. Even though the author provided data showed the infiltration of GZB in tumor tissue is higher than that of control after CAR-T cell treatment, it still can not prove that PET imaging signal is related to the CAR-T cell therapy.
Consequently, the study’s central hypothesis lacks scientific justification. The authors must either (1) provide mechanistic clarification and related references, or (2) rewrite this section to present a more cautious and logical rationale. Otherwise, the manuscript’s conceptual foundation remains untenable.
We thank the reviewer for this thoughtful and scientifically important comment. We agree that immune checkpoint inhibition and CAR-T cell therapy represent mechanistically distinct immunotherapeutic strategies—checkpoint blockade activates endogenous T cells, whereas CAR-T therapy involves adoptively transferred, engineered lymphocytes with different expansion kinetics and trafficking profiles.
Our hypothesis, however, is not intended to imply a direct equivalence between these two modalities, but rather to build on our prior mechanistic findings that 68Ga-NOTA-CYT-200 signal reflects the presence and activity of cytotoxic effector molecules, particularly granzyme B, which is a shared terminal effector of immune-mediated tumor killing across multiple immunotherapeutic contexts. As such, the tracer targets a common downstream biological pathway - cytotoxic granule release by activated effector cells - rather than a therapy-specific upstream mechanism.
In this study, we extend that principle to CAR-T cell–mediated cytotoxicity. Our data demonstrate that:
1.CAR-T–treated tumors show significantly elevated 68Ga-NOTA-CYT-200 uptake compared with controls, consistent with enhanced effector activity.
2.Ex vivo analyses confirm increased intratumoral granzyme B expression and cytotoxic lymphocyte infiltration, supporting the biological link between tracer signal and immune effector function.
3.Prior literature supports the use of granzyme B–targeted PET imaging as a generalized biomarker of cytotoxic T cell activation across various immunotherapies (e.g., Larimer et al., J Nucl Med 2017; Zhou et al., Nat Biomed Eng2020).
We have clarified this conceptual rationale in the revised manuscript (Introduction, paragraph 3) to emphasize that our hypothesis is mechanistically grounded in effector-cell cytotoxicity as a convergent biological endpoint, not in the specific mode of T cell activation. We have adjusted the language of the manuscript to address your concerns about the strength of the conclusions and inferences. We have also added language to the limitations stating that, although we provided evidence of increased cytotoxic T cell activity, we did not directly measure CAR T cell activity in vivo. We hope these changes address your concerns.
And the following comments should be also considered:
1. Please add the description of PET imaging in Introduction part
A detailed description of PET has been added to the Introduction, lines 62-70.
2. All the figures in manuscript and SI should be present in a high-resolution version (e.g., Figure 1,4, and Figure S1)
We uploaded high-resolution versions of all figures contained within the manuscript and the supplementary file.
3. Please add clear scale bar in Figure 4a and give related description in caption part.
We increased the font size of the scale bar and also added a description in the figure legend.
Round 2
Reviewer 1 Report
Comments and Suggestions for Authors
Some conclusions are overstated thus, it is advised to add limitation section.
- Colon uptake remains insufficiently explained, you should put it in limitation section at the end of the discussion (No histology from colon,no cytokine data from colon,no immune-cell staining)
-Also, small sample sizes in PET and cytokine subgroups should be in limitations.
-You should say,, Tracer uptake correlates with GZB activity and CAR T infiltration “ it is more accurate than “Tracer uptake reflects cytotoxic CAR T activity.”
- Some grammar issues: such as PET D7 vs PET D14, immunofluorescent staining…
Author Response
Thank you very much for your thoughtful and constructive feedback, which is sincerely appreciated. We hope that the revisions made in response to your comments have addressed your concerns satisfactorily and have helped to further improve the quality and clarity of the manuscript.
Some conclusions are overstated thus, it is advised to add limitation section.
We agree and have added a dedicated limitations paragraph at the end of the Discussion, where we explicitly qualify the strength of our conclusions and acknowledge key constraints of the study. These changes are highlighted in yellow in the revised manuscript.
- Colon uptake remains insufficiently explained, you should put it in limitation section at the end of the discussion (No histology from colon,no cytokine data from colon,no immune-cell staining)
We agree. The colon uptake findings are now explicitly discussed as an exploratory observation in the Discussion, and the absence of colon histology, colon cytokine data, and colon immune-cell staining has been incorporated into the new limitations paragraph at the end of the Discussion. We have also corrected the wording in the methods section, originally stating that the colon tissue samples were also stained. The corresponding additions are highlighted in yellow.
-Also, small sample sizes in PET and cytokine subgroups should be in limitations.
We agree. The relatively small sample sizes of the PET-imaged cohort and the cytokine-analysis subgroup are now clearly acknowledged in the limitations paragraph of the Discussion. We also briefly note the potential impact on generalizability. These revisions are highlighted in yellow.
-You should say,, Tracer uptake correlates with GZB activity and CAR T infiltration “ it is more accurate than “Tracer uptake reflects cytotoxic CAR T activity.”
We agree and have adjusted the wording accordingly. In the Discussion and Conclusions, we now state that tracer uptake correlates with granzyme B activity and CAR T-cell infiltration, rather than implying that the signal exclusively reflects cytotoxic CAR T activity. All relevant instances have been revised and are highlighted in yellow in the revised manuscript.
- Some grammar issues: such as PET D7 vs PET D14, immunofluorescent staining…
Thank you for pointing this out. We have carefully revised the affected sentences to improve grammar and clarity. PET time-point comparisons (e.g., PET D2 vs PET D7 vs PET D14) have been rewritten for consistency and readability, and the phrases referring to immunohistochemical and immunofluorescent staining have been corrected and streamlined (e.g., “IHC and immunofluorescent staining” / “immunostaining (IHC and immunofluorescence)”). These edits are highlighted in yellow throughout the Methods and Results sections.
We again thank you for your valuable feedback and the time you dedicated to reviewing our work. We believe that your comments have significantly strengthened the manuscript.

Reviewer 3 Report
Comments and Suggestions for Authors
Accept in present form
Author Response
Accept in present form
Thank you very much for your positive assessment of our work. We are delighted that you recommend ‘Accept in present form’ and sincerely appreciate the time and care you devoted to reviewing our manuscript.